# RESOLVING KNOWLEDGE CONFLICTS IN LARGE LANGUAGE MODELS

## ABSTRACT

Large language models (LLMs) often encounter *knowledge conflicts*, scenarios where discrepancy arises between the internal parametric knowledge of LLMs and non-parametric information provided in the prompt context. In this work we ask *what are the desiderata for LLMs when a knowledge conflict arises and whether existing LLMs fulfill them*. We posit that LLMs should 1) *identify knowledge conflicts*, 2) *pinpoint conflicting information segments*, and 3) *provide distinct answers or viewpoints in conflicting scenarios*. To this end, we introduce KNOWL-EDGE CONFLICT, an evaluation framework for simulating contextual knowledge conflicts and quantitatively evaluating to what extent LLMs achieve these goals. KNOWLEDGE CONFLICT includes diverse and complex situations of knowledge conflict, knowledge from diverse entities and domains, two synthetic conflict creation methods, and settings with progressively increasing difficulty to reflect realistic knowledge conflicts. Extensive experiments with the KNOWLEDGE CON-FLICT framework reveal that while LLMs perform well in identifying the existence of knowledge conflicts, they struggle to determine the specific conflicting knowledge and produce a response with distinct answers amidst conflicting information. To address these challenges, we propose new instruction-based approaches that augment LLMs to better achieve the three goals. Further analysis shows that abilities to tackle knowledge conflicts are greatly impacted by factors such as knowledge domain and prompt text, while generating robust responses to knowledge conflict scenarios remains an open research question.

## 1 INTRODUCTION

Large language models (LLMs) have demonstrated remarkable capabilities to encode world knowledge (Peters et al., 2018; Petroni et al., 2019) and solve knowledge-intensive tasks (Roberts et al., 2020; Brown et al., 2020a). Nevertheless, their knowledge abilities are far from perfect (Sun et al., 2023; Hernandez et al., 2023; Muhlgay et al., 2023), leading to the emergence of knowledge augmentation approaches: using external sources (e.g., retrieval corpora (Fisch et al., 2019; Guu et al., 2020; Shi et al., 2023b), search engines (Press et al., 2022; Nakano et al., 2021), and other LMs(Feng et al., 2023d; Luo et al., 2023)) to provide relevant information in the prompt context. However, due to issues such as misinformation, varying perspectives, time-sensitive information, or knowledge updates, **knowledge conflicts** might arise, meaning that there is a discrepancy between *parametric knowledge* (the internal knowledge stored in LLM parameters) and *non-parametric knowledge* (the knowledge fetched from external sources (Chen et al., 2022; Xie et al., 2023)).

Prior research conducted preliminary studies by probing LLMs with knowledge conflicts and examined their behaviors in response (Chen et al., 2022; Li et al., 2023). The key findings are that LLMs' choices between knowledge sources, parametric or non-parametric, depend on factors including the coherence of the external knowledge (Xie et al., 2023) and model size (Longpre et al., 2021). This work extends these prior works by seeking a deeper understanding of whether LLMs can acknowledge knowledge conflicts and how they should respond. Specifically, we ask: *What should be the desirable behaviors of LLMs when knowledge conflicts arise?* and *Are LLMs currently exhibiting those desirable behaviors?* We argue that LLMs should not rely solely on either parametric or non-parametric information, but grant LLM users the agency to make informed decisions based on distinct answers (Floridi, 2023). In line with this goal, we hypothesize that LLMs should 1) identify the existence of knowledge conflicts, 2) pinpoint the specific information segments where

knowledge conflicts occur, and 3) generate distinct responses based on all conflicting information. Achieving these desiderata, as shown in Figure 1, enables LLMs to not only acknowledge the existence of knowledge conflicts but also navigate them skillfully, resulting in responses that are more accurate, comprehensive, and, ultimately, trustworthy.

To this end, we introduce KNOWLEDGE CONFLICT, a framework to simulate contextual knowledge conflicts and evaluate whether LLM's behavior aligns with the three desiderata. Specifically, we first curate a list of 10k entities covering 20 distinct domains and 200 subject areas, while employing two techniques to generate synthetic knowledge conflicts tailored to a specific context. We establish three distinct tasks with increasing complexity to reflect the three goals: 1) *Contextual Knowledge Conflict Detection*: identify the presence of knowledge conflicts, 2) *QA-Span Knowledge Conflict Detection*: determine whether there is a knowledge conflict specifically in a span which is relevant to the question; and 3) *Distinct Answers Generation*: provide distinct answers by leveraging all pieces of conflicting information. These three tasks focus on different aspects of conflict-handling abilities and together serve as a comprehensive evaluation protocol.

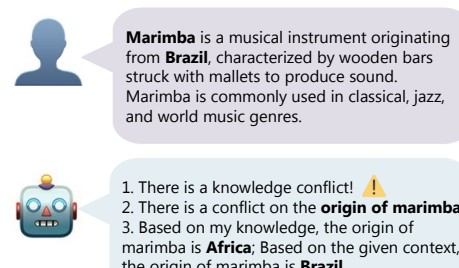

Figure 1: We expect LLMs to 1) acknowledge knowledge conflicts, 2) point out the specific conflicting segments, and 3) generate different answers based on conflicting pieces of information.

We conduct extensive experiments with the KNOWLEDGE CONFLICT framework, revealing that while LLMs perform well above random in Task 1, identifying the existence of knowledge conflicts within contextual information, they encounter notable challenges when it comes to Tasks 2 and 3, which require LLMs to precisely pinpoint these conflicts and provide distinct answers given conflicting context. To address these challenges, we further propose new instruction-based approaches to reflect a wide array of reasoning properties, such as decomposing tasks, breaking down context passages, localization, and more. Through these approaches, we successfully enhance the performance of GPT-3.5-TURBO in Task 1 and Task 3, improving LLM's abilities to acknowledge knowledge conflicts and generate distinct answers amidst conflicting information. Further analyses demonstrate that factors such as knowledge domain, prompt text, and more, while robust handling of knowledge conflicts remains an open research question.

## 2 THE KNOWLEDGE CONFLICT FRAMEWORK

We present the KNOWLEDGE CONFLICT framework, which leverages a wide range of knowledge sources, various conflict creation methods, and progressively challenging settings to reflect real-world knowledge conflicts and assess LLMs' capacity to recognize and address knowledge conflicts. We illustrate the KNOWLEDGE CONFLICT framework in Figure 2.

### 2.1 KNOWLEDGE SCOPE

We generate an entity list as the starting point in KNOWLEDGE CONFLICT by prompting LLMs in a zero-shot manner: we first instruct the LLM to return 20 distinct domains such as `Computer Science`, accompanied by 10 fields within each domain such as `Artificial Intelligence` and `Human-Computer Interaction`, and then 50 entities specific to each field such as `Neural networks` and `User Interface`. As a result, we obtain 9,083 unique entities after filtering out duplicates, covering diverse knowledge areas across various domains. We utilize the generated entity list instead of other publicly accessible entity lists (Pellissier Tanon et al., 2020; Heist & Paulheim, 2020), so it is highly likely that LLMs are familiar with these entities and would contain knowledge and information about them. Note that the KNOWLEDGE CONFLICT framework is independent of the entity list, thus our approach could be easily extended to other domains, subject areas, and more.

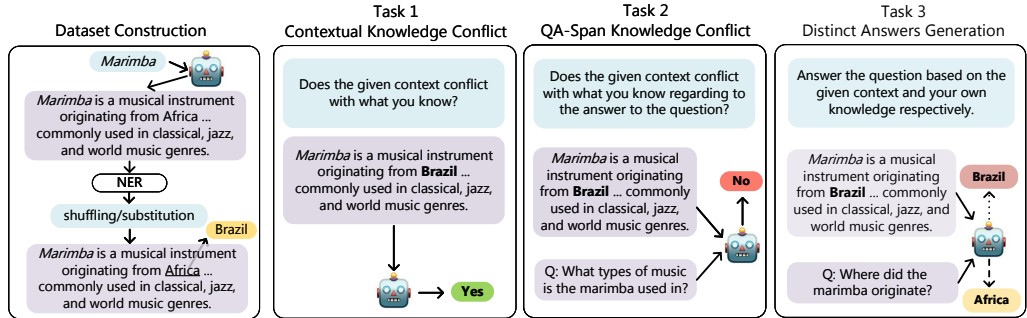

Figure 2: We introduce the KNOWLEDGE CONFLICT framework to comprehensively analyze and improve LLMs' handling of knowledge conflicts. The framework handles concrete spans where knowledge conflicts arise, and facilitates meaningful outputs, granting its users the agency to find appropriate responses in the face of conflicting information.

## 2.2 KNOWLEDGE CONFLICT GENERATION

For each entity, we create two pieces of information by first eliciting the LLM's parametric knowledge about the entity, and then factually modifying it to construct a conflicting knowledge to later put into the prompt context, such that there is a knowledge conflict between these two contexts. We detail the methodology below.

**Parametric Knowledge Elicitation** We instruct LLMs to produce contextual information about each entity under a closed-book setting with the prompt *"Give me some context about {entity} in 50 words."* In this case, LLMs rely solely on their internal parametric knowledge, devoid of external evidence, to generate the requested context. As a result, we adopt the generated context as its parametric knowledge.

**Conflicting Knowledge Creation** We employ two approaches to generate synthetic knowledge conflicts.

- *In-domain Named Entity Substitution*: Inspired by previous works that effectively utilize the entity substitution method (Longpre et al., 2021; Xie et al., 2023), We employ NER models (Honnibal et al., 2020; Liu et al., 2019) to identify named entity categorized as "ordinal", "cardinal", "date", "person", "organization", and "location". We randomly select an identified entity and perform substitution: all occurrences of the selected entity are substituted with another entity of the same type drawn from an in-domain corpus, i.e., an entity of the type "person" will be substituted with another entity in type "person" found in the knowledge contexts generated in parametric knowledge elicitation from the same domain.

- *In-domain Entity Shuffling*: For cases where the NER models fail to identify entities of the six categories, we proceed to shuffle their "main" entities – entities in Section 2.1 that we used to generate these contexts. Concretely, we replace all occurrences of the main entity in the context with another main entity in a context from the same domain.

As a result, both strategies would result in a conflicting passage that conflicts with the parametric knowledge. We further verify this in a human evaluation presented in Section 2.4.

## 2.3 TASKS

After obtaining pairs of passages that are in conflict with each other, we create three tasks to examine LLM's ability to 1) recognize the existence of knowledge conflicts, 2) pinpoint conflicting information segments, and 3) provide different answers to each of the conflicting passages.

**Task 1: Contextual Knowledge Conflict Detection** We set up a binary classification task in which a single piece of context, either its parametric knowledge or the conflicting knowledge, and the instruction *"Does the given context conflict with what you know? Yes/No"* are given in the prompt. The answer *"Yes"* is expected when the conflicting knowledge is given and *"No"* in case of the parametric knowledge. We use Precision, Recall, and F1-score as evaluation metrics.

**Task 2: QA-Span Knowledge Conflict Detection**    It is often the case that not all pieces of information within a passage are in conflict between parametric and conflicting knowledge sources. As a result, in addition to detecting overall contextual knowledge conflict (Task 1), it is crucial for LLMs to pinpoint the specific piece of information where these conflicts arise. We instruct TEXT-DAVINCI-003 (Ouyang et al., 2022) with the prompt *"Given the context, generate a question to which the only single answer is the word {entity} (the question should not contain the word {entity})"*, where the "*entity*" is the entity substituted or shuffled in the conflict generation step, and the conflicting context in a zero-shot manner to generate a question asking about the conflicting segments of the conflicting context. The prompt *"Given the context, generate a question unrelated to {entity}"* is employed to generate a question asking about the non-conflicting segments of the conflicting context. We prompt the LLM with a single context (either parametric knowledge or conflicting knowledge) and a single question (either question about the conflicting segments or question about the non-conflicting segments) with the instruction *"Does the given context conflict with what you know regarding the answer to the question? Yes/No"* for a binary classification. The positive answer is only expected when the conflicting knowledge and the question about the conflicting segments are given. Though we can assess LLMs directly by letting them to identify conflicting segments, we opt for this QA-based method which aligns better with real-world scenarios where users ask questions that might not rely on the conflicting segments. Again, we employ Precision, Recall, and F1-score for evaluation.

**Task 3: Distinct Answers Generation**    While previous studies (Longpre et al., 2021; Xie et al., 2023; Mallen et al., 2023) explored various factors that impact the LLMs to choose between their parametric knowledge and external sources, we believe that it is important to defer agency to the users, i.e., in the face of ambiguity and knowledge conflicts, LLMs should return multiple answers and let users make the choices. Therefore, we test LLMs' ability to generate different answers given conflicting contexts in this task. Specifically, the LLM is given the conflicting text and the question about the conflicting segments of text along with the prompt *"Answer the question based on the given context and your own knowledge respectively."* The ground truths would be the answer based on the conflicting passage and the answer generated by LLMs when only the question is presented in a zero-shot manner. We evaluate the accuracy of parametric-based answers, the accuracy of conflicting-knowledge-based answers, and the accuracy of simultaneously providing both answers.

## 2.4 DATASET ANALYSIS

The dataset we construct through our framework comprises 9,083 distinct entities that are approximately evenly distributed across 20 different domains. Around one-third of the instances of knowledge conflicts stem from named entity substitution, while the remaining two-thirds result from entity shuffling. A detailed breakdown of the dataset by domains and conflict generation methods can be found in Appendix D.

It's worth noting that our conflict generation methods may fail under situations where the context is not unique to a specific entity, for example, when there are multiple individuals holding the title of "chief scientist." To further validate the effectiveness of our conflict generation techniques, we conduct human evaluations for Task 1 and Task 2. Results show that 96% of Task 1 problems and 83.5% of Task 2 problems contain perfectly clear knowledge conflicts. The Fleiss' Kappa (Fleiss, 1971) among the five annotators is 0.51, indicating moderate agreement.

## 3 EXPERIMENT SETTINGS

### 3.1 BASELINES

We evaluate prominent LLM prompting approaches with the KNOWLEDGE CONFLICT framework.

**Zero-shot prompting**    (Liu et al., 2021b) presents LLMs with a problem statement and asks for a direct answer, without any exemplars or intermediate reasoning steps.

**Few-shot prompting**    (Brown et al., 2020b) leverages a few exemplars, pairs of problems and answers, to prompt in-context learning in LLMs.

**Chain-of-Thought prompting (CoT)**    (Wei et al., 2022) includes a reasoning path in in-context exemplars and guides LLMs to follow similar reasoning steps to reach an answer. In Task 1, we

guide LLMs to deconstruct the given context into atomic facts and check if the number of inconsistencies is greater than zero. In Tasks 2 and 3, we lead LLMs to generate the answers based on parametric knowledge and the answers based on the given context separately before the final response.

**Generated Knowledge Prompting (GKP)** (Liu et al., 2021a) involves extracting knowledge from LLMs, and providing it as an additional input when answering a question. We elicit LLMs' parametric knowledge about the main entity in the given context as the supplementary input.

**Self-ask prompting** (Press et al., 2022) requires LLMs to explicitly formulate the next follow-up question they should inquire before answering it. We employ this approach to generate self-ask questions on parametric knowledge and the context provided.

**Break-down prompting** guides LLMs to solve problems or answer questions at the sentence level, and then integrates all responses in the end. We instruct LLMs to perform classification on a sentence-by-sentence basis and then consolidate these individual responses into a coherent answer.

**Self-Consistency (SC)** (Wang et al., 2023b) is a decoding strategy that samples a diverse set of reasoning paths and selects the most consistent answer by marginalizing out the sampled reasoning paths, leveraging the idea that a complex reasoning problem typically admits multiple different ways of thinking leading to its unique correct answer. In our experiments, Self-Consistency is used in conjunction with CoT and GKP.

## 3.2 MODELS AND SETTINGS

We use ChatGPT (Ouyang et al., 2022) (GPT-3.5-TURBO) as the main LLM in the experiments unless otherwise stated. For Self-Consistency that requires multiple samples for a problem, we sample 5 responses with temperature $\tau = 0.7$ following (Wang et al., 2023b); for all the other experiments, we use temperature $\tau = 0$. For few-shot prompting approaches, the input prompt includes four in-context exemplars and their solutions before the problem of interest under Task 1 and Task 2, and two such pairs under Task 3. The in-context exemplars are drawn from different primary fields and different conflict generation methods, and balanced between positive and negative samples.

## 4 RESULTS

**Task 1: Contextual Knowledge Conflict Detection** Table 1 shows that on Task 1, LLMs display a tendency to declare "*no conflict*", which results in a nearly perfect precision but a lower recall. This inclination toward asserting the absence of conflicts can raise doubts about negative predictions, as it doesn't necessarily indicate a genuine assessment of the absence of conflicts. However, these concerns are alleviated when considering the results obtained using CoT, where providing reasons is obligatory. In this scenario, both GKP and Self-ask, methods that rely on explicit classification, do not yield strong performance. This indicates that accurately identifying contextual knowledge conflicts through explicit means is not a trivial task. Overall, the LLM demonstrates an above-random ability in contextual knowledge conflict identification.

| Method | Prec. | Rec. | F1 |
|---|---|---|---|
| ZERO-SHOT | **0.999** | 0.144 | 0.251 |
| FEW-SHOT | **0.999** | 0.351 | 0.520 |
| COT | 0.998 | 0.639 | 0.779 |
| COT + SC | 0.998 | 0.644 | 0.783 |
| GKP + SC | **0.999** | 0.475 | 0.644 |
| SELF-ASK | 0.995 | 0.486 | 0.653 |
| BREAK-DOWN | 0.863 | 0.693 | 0.768 |
| **Ours** | 0.893 | **0.728** | **0.802** |

Table 1: Performance on Task 1: Contextual Knowledge Conflict Detection. The best results are **bold-faced** and the second-best ones are underlined. Our proposed approach outperforms all baselines on F1-score.

**Task 2: QA-Span Knowledge Conflict Detection** Table 2 shows that when the LLM is tasked with the precise identification of knowledge conflicts, its performance experiences a considerable decline. Among all the baseline methods, the self-ask prompting approach stands out as the most effective, and we find that the generated intermediate answers help to narrow down the scope of knowledge conflicts and encourage localization. Also, we observe a consistent pattern where precision exceeds recall. This pattern revalidates LLMs' tendency to assert "*no conflict*". Overall, LLMs struggle to precisely pinpoint the exact piece of information where these conflicts arise.

**Task 3: Distinct Answers Generation** Table 3 shows the results under Task 3 where LLMs are directed to provide answers based on the non-parametric context and its parametric knowledge concurrently. Across all the prompting methods, except for zero-shot where only a single answer is returned in most cases, the accuracy of answers based on the conflicting knowledge surpasses that of parametric-based answers. Break-down prompting is not applicable in this task, and we have omitted Self-Consistency due to the limited improvements it offers in the first two tasks and cost considerations. Overall, the LLM struggles to provide distinct answers simultaneously with the accuracy of getting both correct hovering around 50%, requiring further research and exploration.

| Method | Prec. | Rec. | F1 |
|---|---|---|---|
| ZERO-SHOT | 0.615 | 0.151 | 0.242 |
| FEW-SHOT | 0.395 | **0.860** | 0.541 |
| CoT | 0.843 | 0.375 | 0.519 |
| CoT + SC | 0.875 | 0.367 | 0.517 |
| GKP + SC | 0.508 | 0.499 | 0.504 |
| SELF-ASK | **0.898** | 0.474 | **0.621** |
| BREAK-DOWN | 0.614 | 0.413 | 0.494 |
| **Ours** | 0.718 | 0.426 | 0.535 |

Table 2: Performance on Task 2: QA-Span Knowledge Conflict Detection. The best results are **bold-faced** and the second-best ones are underlined. Self-ask prompting stands out as the strongest baseline method.

## 5 PROPOSED APPROACH

Recent studies (Shi et al., 2023a; Wang et al., 2023a) have shown that instruction-based methods work well to induce new abilities in LLMs. We additionally explore and propose a new set of instruction-based approaches for the three tasks to investigate whether instructions tailored to the context of knowledge conflicts would improve upon generic approaches. Full prompt text is presented in Appendix G.

| Method | Para. | Conflicting. | Both |
|---|---|---|---|
| ZERO-SHOT | 0.400 | 0.350 | 0.031 |
| FEW-SHOT | 0.372 | 0.765 | 0.285 |
| CoT | 0.575 | 0.782 | 0.473 |
| GKP | 0.643 | 0.814 | 0.551 |
| SELF-ASK | 0.611 | 0.735 | 0.464 |
| **Ours** | **0.658** | **0.815** | **0.569** |

Table 3: Performance on Task 3: Distinct Answers Generation. The best results are **bold-faced** and the second-best ones are underlined. Our approach enables LLMs to generate distinct answers supported by different knowledge sources respectively.

**Task 1: Contextual Knowledge Conflict Detection** We propose to employ a four-step approach: 1) elicit knowledge about the main entity, 2) break down the entire context into individual sentences, which draws LLMs' attention to every single detail, and identify sentences that can be verified by the knowledge elicited in step 1, 3) classify whether these sentence conflicts with the knowledge elicited in a sentence-by-sentence manner, and 4) classify whether the remaining sentences conflict with its parametric knowledge (using all the internal knowledge in addition to the knowledge elicited in step 1). For steps 2), 3), and 4), a localization procedure is included, which means apart from returning the answer, the LLMs also need to return their reasoning steps. The main idea is that we promote fine-grained analysis into the sentence-level so that the LLM could better classify and attend to those parts, leaving all the *vague* sentences to the final step. Table 1 shows that our proposed method exhibits a higher F1 score compared to all the baseline methods, albeit with a slight reduction in precision. The efficacy of both the Break-down baseline and our proposed approach underscores that the capacity to discern contextual knowledge conflicts is contingent upon the context's length.

**Task 2: QA-Span Knowledge Conflict Detection** Similarly, we propose to break down the task and fine-grain the context into sentences that can be used to answer the given question. Specifically, the LLMs are instructed to 1) disregard the given context, answer the given question solely based on their own beliefs, 2) find sentences that can be used to answer the given question: if such sentences exist, extract answers from the sentences and determine whether these answers conflict with the answers generated in step 1; if no such sentences exist, report that there is no conflict. As shown in Table 2, Unfortunately, our approach falls short of outperforming all the baseline methods in this setting after great exploration, indicating that instruction-based approaches might be limited in this scenario. This opens up opportunities for future research to enhance the capability of LLMs in pinpointing instances of knowledge conflicts.

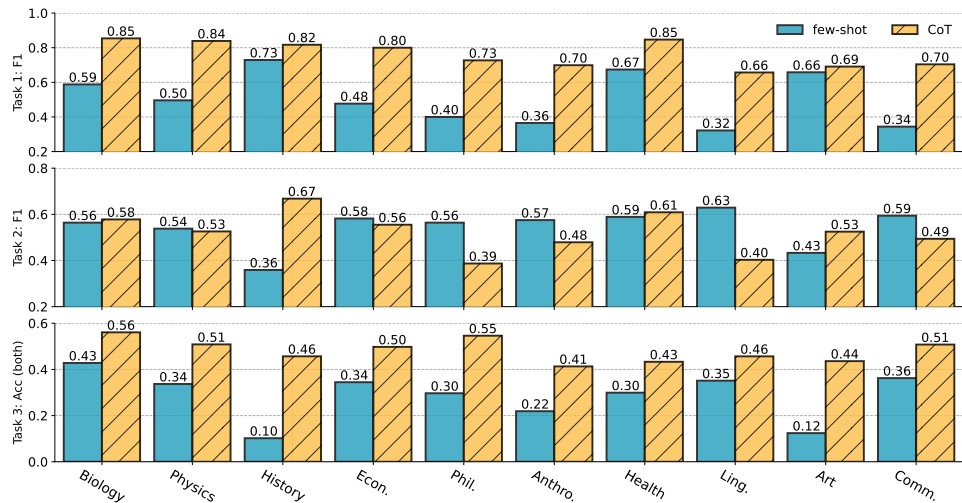

Figure 4: Model performance across knowledge domains on three different tasks. The factor of knowledge domain has a substantial effect on LLM performance on all three tasks, while certain domains (e.g. `art history` and `anthropology`) pose greater challenges.

**Task 3: Distinct Answers Generation**  In order to get more accurate parametric-knowledge-based answers and conflicting-knowledge-based answers, we propose to include "keywords" such as "*solely*" and "*disregard*" to separate the two knowledge sources apart. Also, after generating the response based on one knowledge source, we instruct the LLMs to repeat the question again as LLMs have exhibited limited capability in retaining information across extended contextual spans (Chen et al., 2023; Liu et al., 2023). Table 3 shows that our proposed method attains superior performance across all three metrics.

## 6 ANALYSIS

**Breakdown by Factors**  To examine the factors that may influence the ability of LLMs to identify contextual knowledge conflicts, pinpoint knowledge conflicts, and offer distinct answers when confronted with conflicting information sources, we delve deeper into our results by categorizing them into various domains and conflict generation methods. We also put forward hypotheses regarding the potential reasons for the effects these factors may have.

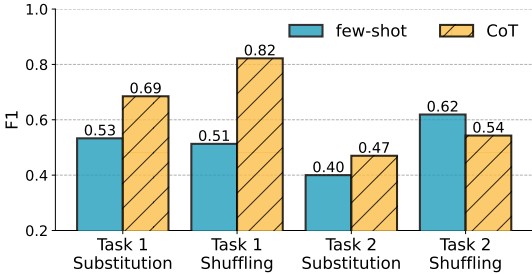

Figure 3: Performance on Tasks 1 and 2 when two conflict creation strategies, named entity substitution and entity shuffling, are separately employed.

• *Knowledge Domain*: As shown in Figure 4, LLMs exhibit a higher proficiency in recognizing (Task 1) and pinpointing (Task 2) contextual knowledge conflict within the domains of `History` and `Health Sciences`. Regarding Task 3, providing distinct answers, LLMs excel in the domains of `Biology`. These results demonstrate that the ability to tackle knowledge conflicts varies across knowledge domains: we hypothesize that it has to do with the quantity of conflicting information present in the pre-training data. Essentially, if LLMs encounter a substantial amount of conflicting information during their pre-training within a specific domain, they tend to perform better in that particular domain. We leave the results for the other 10 knowledge domains in Section Appendix F.

• *Conflict Generation Method*: Figure 3 demonstrates that when we dissect the results according to the two synthetic methods used to generate conflicts (i.e., In-domain Named Entity Substitution and In-domain Entity Shuffling), it becomes evident that LLMs exhibit enhanced performance in scenarios where conflicts are induced through entity shuffling. This outcome aligns with intuition since LLMs find it more straightforward to identify and specify knowledge conflicts when the conflict pertains to the main entity. Results for Task 3 can be found in Appendix F.

| Prompt | Task 1 | | Task 2 | |
|---|---|---|---|---|
| | Few-shot | CoT | Few-shot | CoT |
| Does the given context conflict with what you know...? | 0.529 | 0.805 | 0.541 | 0.546 |
| Does the provided information contradict your existing knowledge...? | 0.529 | 0.795 | 0.541 | 0.580 |
| Is the given information consistent with what you know...? | 0.734 | 0.784 | 0.143 | 0.671 |
| Based on your knowledge, is there anything wrong with the given information...? | 0.616 | 0.810 | 0.442 | 0.628 |
| standard deviation | 0.084 | 0.010 | 0.163 | 0.047 |

Table 5: F1 scores on Tasks 1 and 2 when using different instructions. CoT is more robust than FEW-SHOT in the face of minor changes in instruction texts.

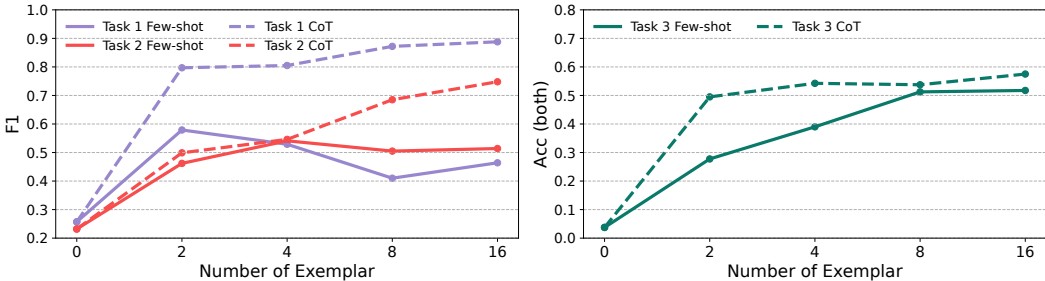

Figure 5: Model performance with various numbers of in-context exemplars across three tasks. CoT benefits more from increasing exemplars, while their impact quickly saturates at 8 examples.

**Prompt Consistency Check** LLMs might be sensitive to subtle changes in the prompt text (Zhu et al., 2023). To further examine the potential impact of instruction phrasing on LLMs' performance in given tasks, we introduce three additional formulations of instructions for each task. Table 5 reveals that, on the whole, performance remains stable across various phrasings. The results also indicate that the CoT prompting method enhances prompt robustness, as evidenced by significantly lower standard deviations in comparison to the Few-shot prompting method. Detailed prompt consistency check results for Task 3 can be found in Appendix F.

| Task | Method | Prec. | Rec. | F1 |
|---|---|---|---|---|
| TASK 1 | fine-tune | 0.977 | 0.930 | 0.953 |
| | prev.best | 0.999 | 0.728 | 0.802 |
| TASK 2 | fine-tune | 0.689 | 0.811 | 0.745 |
| | prev.best | 0.898 | 0.860 | 0.621 |

Table 4: Fine-tuning results on Tasks 1 and 2. While the performance surpasses baselines, it remains short of perfection.

**Number of In-context Exemplars** We explore whether the quantity of in-context exemplars provided affects the ability to tackle knowledge conflicts. Specifically, we change the number of in-context exemplars on a subset of the dataset across the three tasks. As shown in Figure 5, we generally observe that the F1 score plateaus when there are as few as two in-context exemplars. However, including additional exemplars proves beneficial in the case of CoT prompting for Task 2 and Few-shot prompting for Task 3. To sum up, increasing the number of exemplars may improve LLMs' capability of handling knowledge conflicts, but its effect is limited and far from inducing perfectly robust approaches.

**Finetuning** We also finetune the ChatGPT (GPT-3.5-TURBO) under the first and second settings, with training data from the domain of `Political Science` and testing data from the domain of `Geology` to avoid the chance of overfitting. For Task 1, fine-tuning has proven to be exceptionally beneficial, leading to an impressive F1 score of 0.953. However, in the case of Task 2, the impact of fine-tuning is less pronounced, resulting in a comparatively modest F1 score of 0.745. An intriguing observation from Task 2 is that the recall exceeds the precision, which implies a noteworthy reduction in the model's tendency to assert negativity in the context of acknowledging knowledge conflicts.

**More Capable LLMs** We also investigate the competence of other LLMs in tackling knowledge conflicts, which encompass GPT-4 (Bubeck et al., 2023). Due to budget constraints, we only assess its performance on the most challenging Task 3. As an LLM trained on an unprecedented scale of compute and data, GPT-4 showcases increased accuracy in generating both parametric-based answers and conflicting-knowledge-based answers. However, its performance has yet to reach an optimal level, indicating that mere scaling does not solve the challenge of knowledge conflicts.

| Method | Para. | | Conflicting. | | Both | |
|---|---|---|---|---|---|---|
| | GPT-4 | change w/ turbo | GPT-4 | change w/ turbo | GPT-4 | change w/ turbo |
| Zero-shot | 0.206 | -0.194 | 0.628 | +0.278 | 0.057 | +0.026 |
| Few-shot | 0.638 | +0.266 | 0.923 | +0.158 | 0.604 | +0.319 |
| CoT | 0.691 | +0.116 | 0.877 | +0.094 | 0.625 | +0.153 |
| GKP | 0.723 | +0.080 | 0.865 | +0.051 | 0.656 | +0.105 |
| Self-ask | 0.684 | +0.073 | 0.880 | +0.145 | 0.619 | +0.155 |

Table 6: Performance with GPT-4 as the base model and changes with respect to GPT-3.5-TURBO. GPT-4 exhibits improvements across all baselines, while it still falls short of optimal.

## 7 RELATED WORK

**Understanding and expanding the knowledge abilities of LLMs**  Previous works have demonstrated that LLMs have incorporated factual knowledge within their parameters and exhibit considerable potential in recalling factual information (Peters et al., 2018; Petroni et al., 2019; Yu et al., 2023; Mruthyunjaya et al., 2023). However, existing research also reveals that their inherent knowledge is not without flaws (Wu et al., 2022; Pan et al., 2023): outdated knowledge (Hernandez et al., 2023; Yu & Ji, 2023; Padmanabhan et al., 2023), factuality issues (Lee et al., 2022; Feng et al., 2023b), hallucination (Ji et al., 2023; Zhang et al., 2023), and more are common challenges in LLM knowledge abilities. In response, researchers have made concerted efforts to enhance these capabilities through approaches such as retrieval augmentation (Lewis et al., 2020; Guu et al., 2020; Borgeaud et al., 2022; Shi et al., 2023b; Jiang et al., 2023; Zhang et al.), search engine integration (Nakano et al., 2021; Press et al., 2022; Feng et al., 2023a), and incorporating other neural LMs (Feng et al., 2023d; Luo et al., 2023; Du et al., 2023). In this work, we specifically focus on the issue of *knowledge conflict*, when there is a conflict between internal parametric knowledge and external non-parametric knowledge. Without a thorough understanding of how LLMs react to and manage knowledge conflicts, the reliability of their responses may come into question.

**Knowledge Conflict in LLMs**  Previous work on knowledge conflicts primarily focuses on factors impacting models' choice between parametric knowledge and non-parametric knowledge under QA settings. Mallen et al. (2023) finds that conflicting memories are effective for less popular facts; Longpre et al. (2021) explores the effects of model size and retrieval quality by identifying QA instances with named entity answers and substituting mentions of the entity in the gold document with an alternate entity, thus changing the answer; Xie et al. (2023) reveals that when both supportive and contradictory evidence to their parametric memory are present, LLMs show a strong confirmation bias and tend to cling to their parametric memory; Neeman et al. (2023) investigates training data augmentation methods to disentangle parametric and contextual knowledge with counterfactual question answering. Nevertheless, an intriguing and underexplored aspect is to rethink the desiderata for LLMs when confronted with knowledge conflicts, and whether their current responses align with such objectives. To this end, we argue that LLMs should 1) *identify knowledge conflicts*, 2) *pinpoint conflicting information segments*, and 3) *provide distinct answers in conflicting scenarios*. We propose the KNOWLEDGE CONFLICT framework and conduct extensive experiments to evaluate and improve LLMs' ability to tackle knowledge conflicts.

## 8 CONCLUSION

We introduce KNOWLEDGE CONFLICT, an evaluation framework to simulate contextual knowledge conflicts and quantitatively evaluate LLMs' ability to 1) identify contextual knowledge conflicts, 2) pinpoint conflicting knowledge segments, and 3) provide distinct answers or viewpoints amidst conflicts. Extensive experiments demonstrate that LLMs excel at simply identifying knowledge conflicts, but struggle with fine-grained analysis and providing distinct responses. We propose instruction-based approaches that successfully improve in Task 1 and Task 3, while challenges persist in all tasks, especially in Task 2 of pinpointing conflicts. Further analyses show that factors like prompts, exemplars, domains, and conflict simulation methods greatly impact LLM's ability to tackle knowledge conflicts. Our comprehensive framework and in-depth study offer a comprehensive understanding of whether existing LLMs could generate desirable responses amid knowledge conflicts and provide quantitative avenues for evaluating and improving the ability to tackle knowledge conflicts.

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

## A    DISCUSSION AND FUTURE WORK

In our work, we argue that LLMs should be able to acknowledge knowledge conflicts, pinpoint conflicting information segments, and subsequently provide distinct answers grounded on different pieces of conflicting information. This capability not only enhances the robustness of LLMs against malicious attacks by reducing their reliance on a single information source, but also addresses ethical concerns by giving agency to LLM users. Tackling knowledge conflicts in LLMs also holds significance in a socio-political context, as achieving these goals can help LLMs better reflect the plurality of opinions when it comes to social and political issues (Feng et al., 2023c).

In our current task setup, there is one answer derived from the provided context, along with another answer drawn from the parametric knowledge. However, future research can delve into evaluating how well LLMs perform in scenarios where questions can have multiple valid answers stemming from either external information or internal beliefs. Additionally, it would be interesting to explore the possibility of amalgamating the three distinct tasks into one coherent challenge (i.e., instruct LLMs to identify knowledge conflicts, pinpoint knowledge conflicts, and generate distinct answers at the same time) and see if that helps with the performance on individual tasks. Furthermore, it could be intriguing to assess how well LLMs handle perspectives and opinions as conflicting information in addition to facts and knowledge and how the findings might shift with different LLMs, especially those that are open-source.

## B    LIMITATIONS

**Conflicting knowledge may not be factually correct.**    In line with the prior work (Balachandran et al., 2022), we replace facts with a randomly related entity for simplicity. While future work could focus on generating more plausible knowledge alternatives to better align with real-world scenarios, our primary emphasis is the existence of knowledge conflicts instead of factual correctness.

**Assume the top answer as the unique parametric knowledge.**    Throughout this paper, we assume the top answer generated with prompting in a zero-shot manner as the unique *parametric knowledge* of the LLM to the given entity or question (exploration on the possibility of multiple parametric knowledge answers are in Appendix H). However, there is a chance of hallucination in those generated contexts (Ji et al., 2023; Zhang et al., 2023). We argue that since our entity list is generated by probing LLMs, there is a bias towards entities well-known to LLMs, reducing the likelihood of hallucination. The population bias, which is believed to impact whether LLMs rely on parametric knowledge or counter-parametric knowledge (Mallen et al., 2023), is not a significant factor as the LLMs are not instructed to make such a choice in our study. Moreover, there is an ongoing debate about whether hallucination should be considered a form of parametric knowledge, and our focus is not on whether the parametric knowledge is factually accurate or not.

**Real-world knowledge conflicts might be more complex.**    While we employ multiple conflict generation methods and settings to simulate knowledge conflicts, it's crucial to note that our dataset is synthetically generated and limited to word-level edits. Knowledge conflicts found in real-world data might introduce a higher level of complexity to the challenge of knowledge conflicts. For instance, they may involve entirely new entities, increasing the risk of hallucination, while posing new challenges to LLMs and their abilities to tackle knowledge conflicts.

## C    ETHICS STATEMENT

The problem of knowledge conflicts has great ethical implications. In situations where there exists a knowledge conflict, a disparity between external non-parametric knowledge and internal parametric knowledge, there is typically no inherent inclination to favor one over the other. Decisions regarding which knowledge source to trust are context-dependent. For example, in cases necessitating knowledge updates, external knowledge may take precedence. Conversely, in instances involving potential malicious interference, reliance on internal parametric beliefs is advisable. However, when the external information pertains to opinions or diverse perspectives, there may not be an unequivocal correct answer at all. Consequently, navigating knowledge conflicts in such instances raises notable ethical considerations. It is essential to recognize that the baselines and proposed approaches in

| Domain | Substitution | Shuffling | Total |
|---|---|---|---|
| Mathematics | 220 | 273 | 493 |
| Biology | 107 | 337 | 444 |
| Chemistry | 53 | 434 | 487 |
| Physics | 145 | 323 | 468 |
| Psychology | 59 | 377 | 436 |
| Computer Science | 60 | 420 | 480 |
| History | 449 | 2 | 451 |
| Literature | 340 | 137 | 477 |
| Sociology | 41 | 385 | 426 |
| Economics | 67 | 377 | 444 |
| Political Science | 253 | 202 | 455 |
| Philosophy | 108 | 293 | 401 |
| Anthropology | 106 | 338 | 444 |
| Geology | 200 | 260 | 460 |
| Linguistics | 47 | 380 | 427 |
| Art History | 426 | 59 | 485 |
| Environmental Science | 68 | 363 | 431 |
| Health Sciences | 101 | 377 | 478 |
| Communications | 55 | 392 | 447 |
| Music | 175 | 274 | 449 |
| Total | 3,080 | 6,003 | 9,083 |

Table 7: Statistics of our dataset generated with the KNOWLEDGE CONFLICT framework and GPT-3.5-TURBO, separated into knowledge domains and conflict creation strategies.

this study, while valuable for research purposes, are far from perfect and may not perfectly mirror real-world complexities and ethical dilemmas associated with knowledge conflicts.

# D   DATASET DETAILS

We present the detailed breakdown of our dataset by knowledge domains and conflict generation methods in Table 7. Knowledge domains with a higher level of identifiable named entities frequently employ substitution, while others often resort to entity shuffling.

# E   EXPERIMENT DETAILS

Regarding LLMs' responses, we selectively consider only those that not only provide the correct answer but also adhere to the specified response format as true. For Task 3, we take the named entity substituted or the entity shuffled in the given counter-parametric context as the ground truth for counter-parametric-based answers. Regarding ground truth parametric-based answers, we rely on the responses generated by the LLM when only the question is given instead of utilizing the named entity substituted or the entity shuffled in the corresponding parametric context, which helps eliminate cases where multiple answers are possible for the given question based on parametric knowledge. Instances where the ground truth for parametric-based answers coincides with the ground truth for counter-parametric-based answers are removed.

# F   ADDITIONAL ANALYSIS

**Domain Breakdown (cont.)**   In Figure 6, we present the experimental results across the remaining ten knowledge domains on the three tasks. We can see that the ability to identify knowledge conflicts, accurately locate conflicting information, and generate distinct responses based on conflicting data exhibits variability across the various knowledge domains.

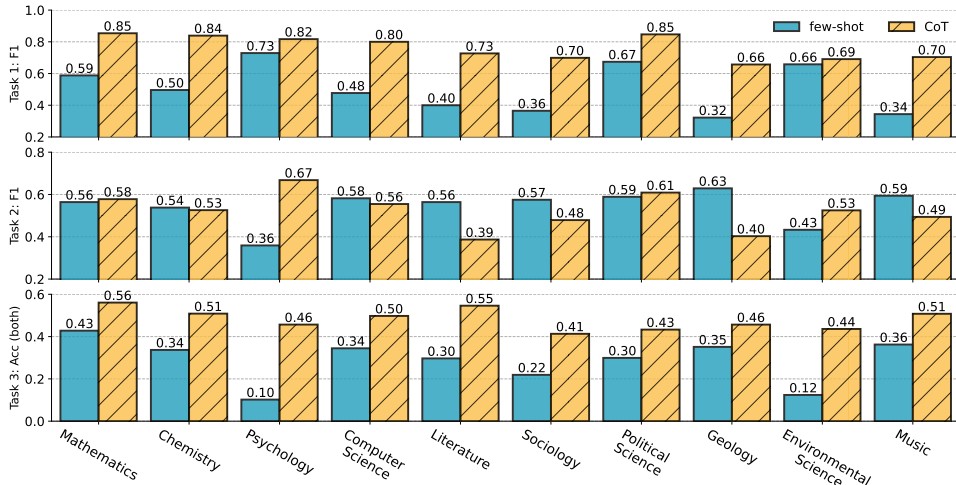

Figure 6: Performance of few-shot and Chain-of-Thought prompting divided into the other ten knowledge domains, following Figure 4.

| Prompt | Few-shot | CoT |
|---|---|---|
| ...based on the given context and your own knowledge respectively. | 0.278 | 0.495 |
| ...considering the provided context and your personal understanding or expertise respectively. | 0.248 | 0.433 |
| Produce two responses...one derived from ... one from ... | 0.420 | 0.483 |
| Produce two responses...one solely based on...one solely based on... | 0.455 | 0.540 |
| **Standard Deviations** | **0.103** | **0.060** |

Table 8: Model performance (Accuracy of both) on Task 3 when different instruction texts are employed.

**Conflict Generation Method Breakdown (cont.)** In Figure 7, we provide a detailed presentation of the results for Task 3, categorized according to the conflict generation methods. These segmented outcomes for Task 3, consistent with the findings for Task 1 and Task 2, reveal that LLMs demonstrate improved performance in situations where conflicts are introduced through entity shuffling.

**Prompt Consistency (cont.)** Furthermore, we carry out a thorough evaluation of prompt consistency in Task 3 by introducing three additional variations of the instruction, as detailed in Table 8. On the whole, the performance remains consistent across these diverse prompts, and the CoT prompting method exhibits a heightened level of prompt robustness.

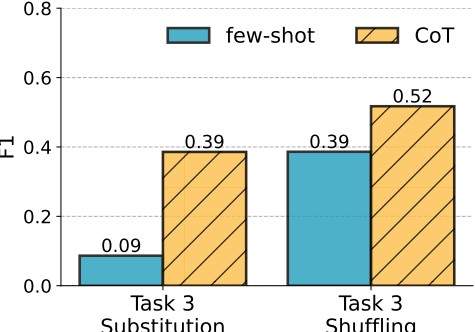

Figure 7: Performance (Accuracy of both) on Tasks 3 when two conflict creation strategies, named entity substitution and entity shuffling, are separately employed, following Figure 3.

**Qualitative Analysis** We take a closer look at the experimental results of a particular example under three settings in this section. In this example, the parametric knowledge is "***Pareto optimality*** *is a concept in economics that refers to a state where no individual can be made better off without making someone else worse off. It represents an efficient allocation of resources, where it is impossible to make any further improvements without negatively impacting someone else's welfare.*" and the conflicting knowledge is "***Trade balance*** *is a concept in economics that refers to a state where no individual can be made better off without making someone else worse off. It represents an efficient allocation of resources, where it is impossible to make any further improvements without negatively impacting someone else's welfare.*" In Table 9 and Table 10, the conflicting knowledge is given in the prompt and the expected response is "*Yes*". In Table

| Method | RESPONSE |
|---|---|
| ZERO-SHOT | No |
| FEW-SHOT | No |
| CoT | Based on my knowledge, Trade balance is a concept in economics that refers to a state where no individual can be made better off without making someone else worse off; Trade balance represents an efficient allocation of resources, where it is impossible to make any further improvements without negatively impacting someone else's welfare. The number of inconsistency is zero, so the answer is No. |
| CoT + SC | Based on my knowledge, Trade balance is a concept in economics that refers to a state where no individual can be made better off without making someone else worse off; Trade balance represents an efficient allocation of resources, where it is impossible to make any further improvements without negatively impacting someone else's welfare. The number of inconsistency is zero, so the answer is No. |
| | Based on my knowledge, trade balance is a concept in economics that refers to a state where no individual can be made better off without making someone else worse off; trade balance represents an efficient allocation of resources, where it is impossible to make any further improvements without negatively impacting someone else's welfare. The number of inconsistency is zero, so the answer is No. |
| | Based on my knowledge, Trade balance is a concept in economics that refers to the difference between the value of a country's exports and the value of its imports; Trade balance represents the state of a country's international trade; Trade balance does not necessarily imply that no individual can be made better off without making someone else worse off; Trade balance does not necessarily represent an efficient allocation of resources. The number of inconsistency is two, which is greater than zero, so the answer is Yes. |
| | Based on my knowledge, Trade balance is a concept in economics that refers to a state where no individual can be made better off without making someone else worse off. It represents an efficient allocation of resources, where it is impossible to make any further improvements without negatively impacting someone else's welfare. The number of inconsistency is zero, so the answer is No. |
| | Based on my knowledge, Trade balance is a concept in economics that refers to a state where no individual can be made better off without making someone else worse off; Trade balance represents an efficient allocation of resources, where it is impossible to make any further improvements without negatively impacting someone else's welfare. The number of inconsistency is zero, so the answer is No. |
| GKP | No |
| | No |
| | No |
| | No |
| | No |

Table 9: Qualitative analysis on Task 1 (Zero-shot, Few-shot, CoT, CoT + SC, and GKP).

11, we provide the conflicting knowledge along with the question "*What is an economic state where no individual can be made better off without making someone else worse off?*" and the expected response is "*Yes*". The same conflicting knowledge and question are also given in Task 3 (Table 12) and the ground truth answers are "*Trade balance*" and "*Pareto efficiency*". We can see that our proposed approach significantly enhances performance.

| Method | RESPONSE |
|---|---|
| SELF-ASK | What does trade balance refer to?
Intermediate answer: Trade balance refers to the difference between a country's exports and imports of goods and services. It is a measure of the net value of a country's trade with the rest of the world.
Follow up: Do these two context directly contradict each other?
Intermediate answer: No. The term "trade balance" is used in both contexts, but they refer to different concepts. In the given context, trade balance refers to an economic concept, while in my knowledge, it refers to a measure of a country's net trade.
So the final answer is: No |
| BREAK-DOWN | There are two sentences in the given context.
Consider the first sentence: Based on what I know, trade balance refers to a state where no individual can be made better off without making someone else worse off. Therefore, there is no conflict in the first sentence.
Consider the second sentence: Based on what I know, trade balance represents an efficient allocation of resources, where it is impossible to make any further improvements without negatively impacting someone else's welfare. Therefore, there is no conflict in the second sentence.
After considering both sentences, there is no conflict. So the final answer is No. |
| OURS | generated knowledge: Trade balance is a term used in economics to describe the difference between the value of a country's exports and the value of its imports. A positive trade balance occurs when a country's exports exceed its imports, resulting in a trade surplus. A negative trade balance occurs when a country's imports exceed its exports, resulting in a trade deficit. The trade balance is an important indicator of a country's economic health and can impact its currency value and overall economic stability.
step 2:
sentence1: Since the generated knowledge includes "Trade balance is a term used in economics to describe the difference between the value of a country's exports and the value of its imports", it can be used to verify the sentence "Trade balance is a concept in economics that refers to a state where no individual can be made better off without making someone else worse off."
step 3:
sentence1: Sentence1 states that trade balance refers to a state where no individual can be made better off without making someone else worse off, and the generated knowledge states that trade balance refers to the difference between the value of a country's exports and the value of its imports, so there is a conflict, yes.
step 4:
remaining sentences: "It represents an efficient allocation of resources, where it is impossible to make any further improvements without negatively impacting someone else's welfare."
Based on what I know, trade balance represents the difference between the value of a country's exports and the value of its imports, and it does not necessarily represent an efficient allocation of resources or the impossibility of making further improvements without negatively impacting someone else's welfare, so there is a conflict, yes. |

Table 10: Qualitative analysis on Task 1 (Self-ask, Break-down, and Ours).

| Method | RESPONSE |
|---|---|
| ZERO-SHOT | No |
| FEW-SHOT | Yes |
| COT | Based on the given context, the answer is trade balance, and based on my knowledge, the answer is also trade balance. So there is no conflict, No. |
| COT + SC | Based on the given context, the answer is trade balance, and based on my knowledge, the answer is also trade balance. So there is no conflict, No.

Based on the given context, the answer is trade balance, and based on my knowledge, the answer is also trade balance. So there is no conflict, No.

Based on the given context, the answer is trade balance, and based on my knowledge, the answer is also trade balance. So there is no conflict, No.

Based on the given context, the answer is trade balance, and based on my knowledge, the answer is also trade balance. So there is no conflict, No.

Based on the given context, the answer is trade balance, and based on my knowledge, the answer is also trade balance. So there is no conflict, No. |
| GKP | No

No

No

No

No |
| SELF-ASK | What is the answer to the given question based on the given context?
Intermediate answer: Based on the given context, the answer is trade balance.
Follow up: What is the answer to the given question based on what you know?
Intermediate answer: Based on what I know, the answer is Pareto efficiency.
Follow up: Do these two answers directly contradict each other?
Intermediate answer: No
So the final answer is: No |
| BREAK-DOWN | There are two sentences in the given context.
Consider the first sentence: The first sentence does contain the answer to the given question, which is trade balance. Based on what I know, the answer is also trade balance. Therefore, there is no conflict.
There is no need to consider further as I already found the sentence containing the answer to the given question. So the final answer is No. |
| OURS | disregard the input, solely based on what I know, the answer is Pareto efficiency
step 2: Since the given question is asking about an economic state where no individual can be made better off without making someone else worse off, the sentence "Trade balance is a concept in economics that refers to a state where no individual can be made better off without making someone else worse off." can be used to answer the given question.
- I find sentences that can be used to answer the given question
    step 3: solely based on the sentence "Trade balance is a concept in economics that refers to a state where no individual can be made better off without making someone else worse off.", the answer is trade balance
    step 4: Since the answer in step 1 is Pareto efficiency and the answer in step 3 is trade balance, there is a conflict, yes. |

Table 11: Qualitative analysis on Task 2.

| Method | RESPONSE |
|---|---|
| ZERO-SHOT | The economic state described in the question is known as a trade balance. |
| FEW-SHOT | Trade balance; Pareto efficiency |
| CoT | Based on the given context, the answer is trade balance; Based on my own knowledge, the answer is Pareto efficiency. |
| GKP | Based on the given context, the answer is trade balance; Based on my own knowledge, the answer is Pareto efficiency. |
| SELF-ASK | What is the answer to the given question based on the given context? Intermediate answer: Based on the given context, the answer is trade balance. Follow up: What is the answer to the given question based on your own knowledge? Intermediate answer: Based on my own knowledge, the answer is Pareto efficiency. So the final answer is: Based on the given context, the answer is trade balance; Based on my own knowledge, the answer is Pareto efficiency. |
| OURS | In the given context, trade balance is an economic state where no individual can be made better off without making someone else worse off, so ANSWER1 is trade balance. Now disregard the given context and answer the given question "What is an economic state where no individual can be made better off without making someone else worse off?" solely based on your own knowledge. Answer: Solely based on my own knowledge, the economic state where no individual can be made better off without making someone else worse off is called Pareto efficiency, so ANSWER2 is Pareto efficiency. |

Table 12: Qualitative analysis on Task 3.

## G  PROMPT TEXT

We present all adopted prompt texts in Table 13 to 20 to facilitate reproducibility.

## H  MULTIPLE PARAMETRIC KNOWLEDGE ANSWERS

We delve deeper into the possibility of multiple parametric knowledge responses, relaxing the assumption of a singular parametric knowledge answer in the main experiments, and examine how the results may be altered. We sample 600 questions (with conflicts) across different knowledge domains, and employ GPT-3.5-TURBO to generate 3 answers in a zero-shot manner with a temperature of 1.5 to try to induce multiple parametric knowledge answers, following self-consistency (Wang et al., 2023b). We present the results in Table 21.The results show that 25% of the additional parametric answers matched the top answer exactly with 65% of synonyms, while others tend to be invalid or incorrect answers. Therefore, we conclude that the vast majority of questions have only one parametric answer, thus the results and findings won't change significantly if we relax this assumption.

## I  MULTI-HOP QUESTIONS

Given the prevalence of multi-hop questions in practical scenarios and the difficulty of pinpointing the step where knowledge conflicts occur, we extend our investigation to assess the ability of identifying knowledge conflicts using a two-hop dataset, Compositional Celebrities (Press et al., 2022). We randomly choose 400 examples and replace the LLM responses to one of the sub-questions with a different answer from the in-context corpus, creating 400 positive examples and 400 negative examples. Employing the CoT prompting method (Wei et al., 2022) results in a precision of 0.795, a recall of 0.995, and an F1 score of 0.884 in Task 1 22. This aligns with our conclusion that LLMs can perform well above random in identifying the existence of knowledge conflicts within contextual information, even under multi-hop settings.

| Method | PROMPT |
|---|---|
| ZERO-SHOT | {context}
Does the information above conflict with what you know? Yes/No |
| Few-shot | Does the given context conflict with what you know? Yes/No

Examples:
Context: Acetone is an organic compound and the simplest amide derived from acetic acid. It is a white crystalline solid that is soluble in water and commonly used as a solvent in laboratories. It is also utilized in the production of pharmaceuticals, dyes, and plastics.
Answer: Yes

Context: {context}
Answer: |
| CoT | Does the given context conflict with what you know? Yes/No

Examples:
Context: Acetone is an organic compound and the simplest amide derived from acetic acid. It is a white crystalline solid that is soluble in water and commonly used as a solvent in laboratories. It is also utilized in the production of pharmaceuticals, dyes, and plastics.
Answer: Based on my knowledge, Acetone is an organic compound; Acetone is a type of ketone instead of amide; Acetone is a colorless liquid instead of white solid; Acetone is soluble in water; Acetone is commonly used as a solvent in laboratories; Acetone is utilized in the production of acrylic plastics, polycarbonates and epoxy resins instead of dyes. The number of inconsistency is three, which is greater than zero, so the answer is Yes.

Context: {context}
Answer: |
| GKP (KNOWLEDGE GENERATION) | Generate some knowledge about the main entity in the input. Examples:

Input: Acetamide is an organic compound and the simplest amide derived from acetic acid. It is a white crystalline solid that is soluble in water and commonly used as a solvent in laboratories. It is also utilized in the production of pharmaceuticals, dyes, and plastics. Knowledge: Acetamide is a organic compound with the chemical formula $CH_3CONH_2$. It is a white, crystalline solid and is the simplest amide derived from acetic acid. Acetamide is commonly used in laboratory settings as a solvent, and it also has applications in various industries. It can be used in the production of pharmaceuticals, plastics, and as a precursor to other organic compounds.

Input: {context}
Knowledge: |
| GKP (KNOWLEDGE INTEGRATION) | Does the given context conflict with what you know? (The given knowledge from you may help with the judgment) Yes/No

Examples:
Context: Acetamide is an organic compound and the simplest amide derived from acetic acid. It is a white crystalline solid that is soluble in water and commonly used as a solvent in laboratories. It is also utilized in the production of pharmaceuticals, dyes, and plastics. Knowledge: Acetamide is a organic compound with the chemical formula $CH_3CONH_2$. It is a white, crystalline solid and is the simplest amide derived from acetic acid. Acetamide is commonly used in laboratory settings as a solvent, and it also has applications in various industries. It can be used in the production of pharmaceuticals, plastics, and as a precursor to other organic compounds.
Answer: No

Context: {context}
Knowledge: {knowledge}
Answer: |

Table 13: Prompts for Task 1 (Zero-shot, Few-shot, CoT, and GKP). The placeholder {context} is substituted for either parametric knowledge or conflicting knowledge at the test time, while the placeholder {knowledge} is substituted for the knowledge generated in the GKP knowledge generation step. The number of exemplars in the table may be less than real.

| Method | PROMPT |
|---|---|
| SELF-ASK | Does the given context conflict with what you know? Yes/No Examples:

Context: Acetone is an organic compound and the simplest amide derived from acetic acid. It is a white crystalline solid that is soluble in water and commonly used as a solvent in laboratories. It is also utilized in the production of pharmaceuticals, dyes, and plastics.
Follow up: What is acetone?
Intermediate answer: Acetone is a colorless, flammable liquid solvent widely used in industry and households. It's known for its strong, sweet odor and high volatility. Acetone is utilized as a solvent in paints, varnishes, nail polish removers, and as a key ingredient in the production of plastics, fibers, and pharmaceuticals.
Follow up: Do these two context directly contradict each other?
Intermediate answer: Yes. In the given context, acetone is a white crystalline solid, while it is a colorless liquid based on what I know.
So the final answer is: Yes

Context: {context}
Follow up: |
| BREAK-DOWN | Does any of the sentence in the given context conflict with what you know? Yes/No Examples:

Context: Acetamide is an organic compound and the simplest amide derived from acetic acid. It is a white crystalline solid that is soluble in water and commonly used as a solvent in laboratories. It is also utilized in the production of pharmaceuticals, dyes, and plastics.
Answer: There are three sentences in the given context.
Consider the first sentence: Based on what I know, acetamide is an organic compound and the simplest amide derived from acetic acid. Therefore, there is no conflict in the first sentence.
Consider the second sentence: Based on what I know, acetamide is a white crystalline solid that is soluble in water and commonly used as a solvent in laboratories. Therefore, there is no conflict in the second sentence.
Consider the third sentence: Based on what I know, acetamide is utilized in the production of pharmaceuticals, dyes, and plastics. Therefore, there is no conflict in the third sentence.
After considering all three sentences, there is no conflict. So the final answer is No.

Context: {context}
Answer: |

Table 14: Prompts for Task 1 (Self-ask and Break-down). The placeholder {context} is substituted for either parametric knowledge or conflicting knowledge at the test time. The number of exemplars in the table may be less than real.

| Method | PROMPT |
|--------|--------|
| OURS | Given the input:
step 1: generate some knowledge about the main entity in the input.
step 2: from the input, find sentences that can be verified by the generated knowledge
step 3: for each sentence found in step2, does it conflict with the generated knowledge? Yes/No
step 4: for the remaining sentences that cannot be verified by the generated knowledge, does it conflict with what you know?

Input: Acetone is an organic compound and the simplest amide derived from acetic acid. It is a white crystalline solid that is soluble in water and commonly used as a solvent in laboratories. It is also utilized in the production of pharmaceuticals, dyes, and plastics.
step 1:
generated knowledge: Acetone is a colorless, flammable liquid widely used in industry and households. It's known for its strong, sweet odor and high volatility. Acetone is utilized as a solvent in paints, varnishes, nail polish removers, and as a key ingredient in the production of plastics, fibers, and pharmaceuticals.
step 2:
sentence1: Since the generated knowledge includes "Acetone is a colorless, flammable liquid widely used in industry and households.", it can be used to verify the sentence "It is a white crystalline solid that is soluble in water and commonly used as a solvent in laboratories."
sentence2: Since the generated knowledge includes "Acetone is utilized as a solvent in paints, varnishes, nail polish removers, and as a key ingredient in the production of plastics, fibers, and pharmaceuticals.", it can be used to verify the sentence "It is also utilized in the production of pharmaceuticals, dyes, and plastics."
step 3:
sentence1: Sentence1 states that acetone is a while crystalline solid, and the generated knowledge states that acetone is a colorless liquid, so there is a conflict, yes.
sentence2: Sentence2 states that acetone can be utilized in the production of pharmaceuticals, dyes, and plastics, and the generated knowledge states that acetone is utilized as a solvent in paints and as a key ingredient in the production of plastics and pharmaceuticals, so there is no conflict, no.
step 4:
remaining sentences: "Acetone is an organic compound and the simplest amide derived from acetic acid."
Based on what I know, acetone is an organic compound, but it is not an amide, so there is a conflict, yes.

Input: {context}
step 1: |

Table 15: Prompts for Task 1 (Ours). The placeholder {context} is substituted for either parametric knowledge or conflicting knowledge at the test time. The number of exemplars in the table may be less than real.

| Method | PROMPT |
|---|---|
| ZERO-SHOT | {context}
Does the information above conflict with what you know regarding the answer to the question {question}? Yes/No |
| FEW-SHOT | Does the given context conflict with what you know regarding the answer to the question? Yes/No Examples:

Context: Acetone is an organic compound and the simplest amide derived from acetic acid. It is a white crystalline solid that is soluble in water and commonly used as a solvent in laboratories. It is also utilized in the production of pharmaceuticals, dyes, and plastics. Question: What is the chemical name of the simplest amide derived from acetic acid? Answer: Yes

Context: {context}
Question: {question}
Answer: |
| COT | Does the given context conflict with what you know regarding the answer to the question? Yes/No

Examples:
Context: Acetone is an organic compound and the simplest amide derived from acetic acid. It is a white crystalline solid that is soluble in water and commonly used as a solvent in laboratories. It is also utilized in the production of pharmaceuticals, dyes, and plastics. Question: What is the chemical name of the simplest amide derived from acetic acid? Answer: Based on the given context, the answer is acetone, while based on my knowledge, the answer is acetamide. So there is a conflict, Yes.

Context: {context}
Question: {question}
Answer: |
| GKP (KNOWLEDGE GENERATION) | Generate some knowledge about the input. Examples:

Input: What is the chemical name of the simplest amide derived from acetic acid?
Knowledge: Acetamide is the chemical name of the simplest amide derived from acetic acid.

Input: {question}
Knowledge: |
| GKP (KNOWLEDGE INTEGRATION) | Does the given context conflict with what you know regarding the answer to the given question? (The given knowledge from you may help with the judgment)

Examples:
Context: Acetone is an organic compound and the simplest amide derived from acetic acid. It is a white crystalline solid that is soluble in water and commonly used as a solvent in laboratories. It is also utilized in the production of pharmaceuticals, dyes, and plastics. Question: What is the chemical name of the simplest amide derived from acetic acid? Knowledge: Acetamide is the chemical name of the simplest amide derived from acetic acid. Answer: Yes

Context: {context}
Question: {question}
Knowledge: {knowledge}
Answer: |

Table 16: Prompts for Task 2 (Zero-shot, Few-shot, CoT, and GKP). At test time, the placeholder {context} is interchanged with either parametric knowledge or conflicting knowledge, the placeholder {question} is replaced by either the question concerning conflicting segments or the question regarding non-conflicting segments, and {knowledge} is substituted with the knowledge produced during the GKP knowledge generation step. The number of exemplars in the table may be less than real.

| Method | PROMPT |
|---|---|
| SELF-ASK | Does the given context conflict with what you know regarding the answer to the given question? Yes/No Examples: |
| | Context: Acetone is an organic compound and the simplest amide derived from acetic acid. It is a white crystalline solid that is soluble in water and commonly used as a solvent in laboratories. It is also utilized in the production of pharmaceuticals, dyes, and plastics. Question: What is the chemical name of the simplest amide derived from acetic acid? Follow up: What is the answer to the given question based on the given context? Intermediate answer: Based on the given context, the answer is acetone. Follow up: What is the answer to the given question based on what you know? Intermediate answer: Based on what I know, the answer is acetamide. Follow up: Do these two answers directly contradict each other? Intermediate answer: Yes So the final answer is: Yes |
| | Context: {context} Question: {question} Follow up: |

Table 17: Prompts for Task 2 (Self-ask). At test time, the placeholder {context} is interchanged with either parametric knowledge or conflicting knowledge and the placeholder {question} is replaced by either the question concerning conflicting segments or the question regarding non-conflicting segments. The number of exemplars in the table may be less than real.

| BREAK-DOWN | Does any of the sentence in the given context conflict with what you know regarding the answer to the given question? Yes/No Examples: 

 Context: Acetone is an organic compound and the simplest amide derived from acetic acid. It is a white crystalline solid that is soluble in water and commonly used as a solvent in laboratories. It is also utilized in the production of pharmaceuticals, dyes, and plastics. 
 Question: What is the chemical name of the simplest amide derived from acetic acid? 
 Answer: There are three sentences in the given context. 
 Consider the first sentence: The first sentence does contain the answer to the given question, which is acetone. Based on what I know, the answer is acetamide. Therefore, there is a conflict. 
 There is no need to consider further as I already found the sentence containing the answer to the given question. So the final answer is Yes. 

 Context: {context} 
 Question: {question} 
 Answer: |
|---|---|
| OURS | Given the input and the question: 
 step 1: disregard the input, answer the given question solely based on what you know 
 step 2: from the input, find sentences that can be used to answer the given question if any 
 - if you find sentences that can be used to answer the given question: 
 step 3: extract the answer from these sentences 
 step 4: does the answer in step 1 conflict with the answer in step 3? Yes/No 
 - if you don't find sentences that can be used to answer the given question: 
 step 3: return there is no conflict, no. 

 Input: Acetone is an organic compound and the simplest amide derived from acetic acid. It is a white crystalline solid that is soluble in water and commonly used as a solvent in laboratories. It is also utilized in the production of pharmaceuticals, dyes, and plastics. 
 Question: What is the chemical name of the simplest amide derived from acetic acid? 
 step 1: disregard the input, solely based on what I know, the answer is acetamide 
 step 2: Since the given question is asking about the chemical name of the simplest amide derived from acetic acid, the sentence "Acetone is an organic compound and the simplest amide derived from acetic acid." can be used to answer the given question. 
 - I find sentences that can be used to answer the given question 
 step 3: solely based on the sentence "Acetone is an organic compound and the simplest amide derived from acetic acid.", the answer is acetone 
 step 4: Since the answer in step 1 is acetamide, and the answer in step 3 is acetone, there is a conflict, yes. 

 Input: {context} 
 Question: {question} 
 step 1: |

Table 18: Prompts for Task 2 (Break-down and Ours). The placeholder {context} is substituted for either parametric knowledge or conflicting knowledge, while the placeholder {question} is substituted for either the question about the conflicting segments or the question about the non-conflicting segments at the test time. The number of exemplars in the table may be less than real.

| Method | PROMPT |
|---|---|
| ZERO-SHOT | Context: {context}
Question: {question}
Answer the given question based on the given context and your own knowledge respectively.
Answer: |
| FEW-SHOT | Answer the question based on the given context and your own knowledge respectively. Examples:

Context: Acetone is an organic compound and the simplest amide derived from acetic acid. It is a white crystalline solid that is soluble in water and commonly used as a solvent in laboratories. It is also utilized in the production of pharmaceuticals, dyes, and plastics.
Question: What is the chemical name of the simplest amide derived from acetic acid?
Answer: acetone; acetamide

Context: {context}
Question: {question}
Answer: |
| CoT | Answer the question based on the given context and your own knowledge respectively. Examples:

Context: Acetone is an organic compound and the simplest amide derived from acetic acid. It is a white crystalline solid that is soluble in water and commonly used as a solvent in laboratories. It is also utilized in the production of pharmaceuticals, dyes, and plastics.
Question: What is the chemical name of the simplest amide derived from acetic acid?
Answer: Based on the given context, the answer is acetone; Based on my own knowledge, the answer is acetamide.

Context: {context}
Question: {question}
Answer: |
| GKP | Answer the question based on the given context and your own knowledge respectively. (You may find the given knowledge from you helpful.) Examples:

Context: Acetone is an organic compound and the simplest amide derived from acetic acid. It is a white crystalline solid that is soluble in water and commonly used as a solvent in laboratories. It is also utilized in the production of pharmaceuticals, dyes, and plastics.
Question: What is the chemical name of the simplest amide derived from acetic acid?
Knowledge: Acetamide is the chemical name of the simplest amide derived from acetic acid.
Answer: Based on the given context, the answer is acetone; Based on my own knowledge, the answer is acetamide.

Context: {context}
Question: {question}
Knowledge: {knowledge}
Answer: |

Table 19: Prompts for Task 3 (Zero-shot, Few-shot, CoT, and GKP). During the test phase, the placeholder {context} is exchanged with either parametric knowledge or conflicting knowledge, while the placeholder {question} is replaced with either the question pertaining to conflicting segments or the question regarding non-conflicting segments. Similarly, {knowledge} is replaced with the knowledge that is generated, as in Task 2. The number of exemplars in the table may be less than real.

| Method | PROMPT |
|---|---|
| SELF-ASK | Answer the question based on the given context and your own knowledge respectively. Examples:

Context: Acetone is an organic compound and the simplest amide derived from acetic acid. It is a white crystalline solid that is soluble in water and commonly used as a solvent in laboratories. It is also utilized in the production of pharmaceuticals, dyes, and plastics.
Question: What is the chemical name of the simplest amide derived from acetic acid?
Follow up: What is the answer to the given question based on the given context?
Intermediate answer: Based on the given context, the answer is acetone.
Follow up: What is the answer to the given question based on your own knowledge?
Intermediate answer: Based on my own knowledge, the answer is acetamide.
So the final answer is: Based on the given context, the answer is acetone; Based on my own knowledge, the answer is acetamide.

Context: {context}
Question: {question}
Follow up: |
| OURS | Generate two answers to the given question: ANSWER1 solely based on the given context and ANSWER2 solely based on your own knowledge. Examples:

Question: What is the chemical name of the simplest amide derived from acetic acid?
Answer the question solely based on the given context: Acetone is an organic compound and the simplest amide derived from acetic acid. It is a white crystalline solid that is soluble in water and commonly used as a solvent in laboratories. It is also utilized in the production of pharmaceuticals, dyes, and plastics.
Answer: In the given context, acetone is the simplest amide derived from acetic acid, so ANSWER1 is acetone.
Now disregard the given context and answer the given question "What is the chemical name of the simplest amide derived from acetic acid?" solely based on your own knowledge.
Answer: Solely based on my own knowledge, the chemical name of the simplest amide derived from acetic acid is acetamide, so ANSWER2 is acetamide.

Question: {question}
Answer the question solely based on the given context: {context}
Answer: |

Table 20: Prompts for Task 3 (Self-ask and Ours). During the test phase, the placeholder {context} is exchanged with either parametric knowledge or conflicting knowledge, while the placeholder {question} is replaced with either the question pertaining to conflicting segments or the question regarding non-conflicting segments. The number of exemplars in the table may be less than real.

| Domain | ExactMatch | Synonyms | Invalid/Incorrect |
|--------|------------|----------|-------------------|
| Mathematics | 8 | 17 | 5 |
| Biology | 9 | 18 | 3 |
| Chemistry | 7 | 20 | 3 |
| Physics | 4 | 23 | 3 |
| Psychology | 6 | 20 | 4 |
| Computer Science | 7 | 21 | 2 |
| History | 11 | 18 | 1 |
| Literature | 15 | 14 | 1 |
| Sociology | 3 | 27 | 0 |
| Economics | 9 | 19 | 2 |
| Political Science | 10 | 16 | 4 |
| Philosophy | 5 | 21 | 4 |
| Anthropology | 6 | 24 | 0 |
| Geology | 8 | 17 | 5 |
| Linguistics | 1 | 25 | 4 |
| Art History | 12 | 10 | 8 |
| Environmental Science | 8 | 19 | 3 |
| Health Sciences | 7 | 22 | 1 |
| Communications | 9 | 20 | 1 |
| Music | 3 | 20 | 7 |
| **Total** | **148** | **391** | **61** |

Table 21: Consistency of 3 parametric knowledge answers generated by GPT-3.5-TURBO on 600 randomly sampled questions. The majority of additional answers are found to be nearly the same, showing that, without the assumption of unique parametric knowledge answer, our conclusions won't change significantly.

| Method | n | TP | TN | FP | FN | Acc | Precision | Recall | F1 |
|--------|---|----|----|----|----|-----|-----------|--------|-----|
| ZERO-SHOT | 743 | 147 | 298 | 73 | 225 | 0.599 | 0.668 | 0.395 | 0.497 |
| FEW-SHOT | 796 | 164 | 224 | 174 | 234 | 0.487 | 0.485 | 0.412 | 0.446 |
| CoT | 756 | 373 | 285 | 96 | 2 | 0.870 | 0.795 | 0.995 | 0.884 |

Table 22: Results of the Compositional Celebrities (Press et al., 2022) dataset on Task 1, which aligns with our conclusion that LLMs exhibit proficiency beyond random chance in detecting the presence of knowledge conflicts.

