# OpenReview forum: "Resolving Knowledge Conflicts in Large Language Models"
_ICLR.cc/2024/Conference — Submitted to ICLR 2024_

### Official Review · Reviewer_PtZ2 · 2023-10-27

**Soundness:** 3 good
**Presentation:** 1 poor
**Contribution:** 3 good
**Rating:** 6
**Confidence:** 3

**Summary:**

In this work, the authors introduce KNOWLEDGE CONFLICT, an evaluation framework for simulating contextual knowledge
conflicts and quantitatively evaluating to what extent LLMs achieve the following goals: 1) identify knowledge conflicts, 2) pinpoint conflicting information segments, and 3) provide distinct answers or viewpoints in conflicting scenarios.

**Strengths:**

1)  The authors design a series of knowledge conflict tasks to measure the performance of existing LLMs to generate response based on conflicted knowledge
2）The developed framework is technically sound and easy to follow

**Weaknesses:**

The most shorting is the orginize of the whole article and detailed questions can be found in the following part

**Questions:**

First of all, I really apprecitate the author start to investigate the problem of knowlege confilct in LLMs, which is urgent need to be solved in practical Chatbot. And the authors also developed a knowledge confilct framework equipped with a series of tasks to solve this question.


However, the originize of this paper is really poor, the authors mixed the design of each taskd and their method in the same part, which will confuse the readers. I really suggest the authors to pay more attention to the writting of this paper.

For technical details, in Task 1, I wonder which kind of pomprt you used in other baselines, like Few-shot prompting and Chain-of-Thought prompting (CoT), and your method in Table.1 indicates using the prompt ``Does the given context conflict with what you know? Yes/No''.
Then, for each experimental result, I wonder if your method will include some examplers as demonstrations or just the prompt mentioned in your article?

Further, for NER in the first step, I wonder how you deal with the senario of there are multiple entity term in a sentence, will you enumerate them and generate context of parmetric knowledge for them?

Overall, the method is simple but will be useful in my understanding, but I wonder author have idea to how to combine your method with existing prompting method, like CoT, because there are still a lot of multi-hop question in practice and you will not know the knowledge confilct happens in which step, just like when you solve a math problem, you don't which confictled knowledge leads to a wrong intermediate result

---

> ### Author Response · Authors · 2023-11-16
>
> We would like to thank the reviewer for their thoughtful comments and feedback. Their positive feedback regarding our design of knowledge conflict tasks and framework, as well as the overall idea is motivating for us.
>
> >First of all, I really apprecitate the author start to investigate the problem of knowlege confilct in LLMs, which is urgent need to be solved in practical Chatbot. And the authors also developed a knowledge confilct framework equipped with a series of tasks to solve this question.
>
> Thank you! We appreciate your positive feedback!
>
> > ​​However, the originize of this paper is really poor, the authors mixed the design of each taskd and their method in the same part, which will confuse the readers. I really suggest the authors to pay more attention to the writting of this paper.
>
> Thanks for your suggestion. Though we believe that grouping by tasks and grouping by content sections are both valid writing strategies, and we employ grouping by sections (settings, results, and proposed approaches) with subtitles of "Task1", "Task2", and "Task3" in each of them to help with the comparison on methodologies and results across different tasks, similar to other works on knowledge conflicts [1-2], we will clarify better the organization of the paper and emphasize on the separation of the conceptual design from the method in the final version.
>
> > For technical details, in Task 1, I wonder which kind of pomprt you used in other baselines, like Few-shot prompting and Chain-of-Thought prompting (CoT), and your method in Table.1 indicates using the prompt ``Does the given context conflict with what you know? Yes/No''. Then, for each experimental result, I wonder if your method will include some examplers as demonstrations or just the prompt mentioned in your article?
>
> We present all prompts in Appendix G. To reiterate:
>
> For the few-shot prompting, we used "*Does the given context conflict with what you know? Yes/No Examples: Context: Acetone is an organic compound and the simplest amide derived from acetic acid. It is a white crystalline solid that is soluble in water and commonly used as a solvent in laboratories. It is also utilized in the production of pharmaceuticals, dyes, and plastics. Answer: Yes Context: {context} Answer:*".
>
> For chain-of-thought prompting, we used "*Does the given context conflict with what you know? Yes/No Examples: Context: Acetone is an organic compound and the simplest amide derived from acetic acid. It is a white crystalline solid that is soluble in water and commonly used as a solvent in laboratories. It is also utilized in the production of pharmaceuticals, dyes, and plastics. Answer: Based on my knowledge, Acetone is an organic compound; Acetone is a type of ketone instead of amide; Acetone is a colorless liquid instead of white solid; Acetone is soluble in water; Acetone is commonly used as a solvent in laboratories; Acetone is utilized in the production of acrylic plastics, polycarbonates and epoxy resins instead of dyes. The number of inconsistency is three, which is greater than zero, so the answer is Yes. Context: {context} Answer:*``.
>
> Four in-context exemplars are included for few-shot prompting approaches (few-shot, CoT, GKP, self-ask, break-down, and ours) as documented in Section 3.2.
>
> [1] Longpre, Shayne et al. Entity-based knowledge conflicts in question answering. EMNLP 2021.
>
> [2] Neeman Ella et al. DisentQA: Disentangling Parametric and Contextual Knowledge with Counterfactual Question Answering. ACL 2023.

---

> ### Author Response · Authors · 2023-11-16
>
> > Further, for NER in the first step, I wonder how you deal with the senario of there are multiple entity term in a sentence, will you enumerate them and generate context of parmetric knowledge for them?
>
> As stated in Section 2.2, we randomly select one identified entity for each context instead of enumerating all entities on a smaller set of contexts to obtain broader coverage.
>
> > Overall, the method is simple but will be useful in my understanding, but I wonder author have idea to how to combine your method with existing prompting method, like CoT, because there are still a lot of multi-hop question in practice and you will not know the knowledge confilct happens in which step, just like when you solve a math problem, you don't which confictled knowledge leads to a wrong intermediate result
>
> Thank you for acknowledging the usefulness of our method and for your relevant question! Indeed, our proposed approaches (as fully documented in Appendix G) do incorporate existing prompting methods, including the CoT.
>
> Regarding your concern on multi-hop questions, we conduct **an additional experiment** utilizing the two-hop dataset Compositional Celebrities [1]. We randomly choose 400 examples and replace the LLM responses to one of the sub-questions with a different answer from the in-context corpus, creating 400 positive examples and 400 negative examples. Employing the CoT prompting method results in a precision of 0.795, a recall of 0.995, and an F1 score of 0.884 in Task 1. This aligns with our conclusion that LLMs can perform well above random in identifying the existence of knowledge conflicts within contextual information, even under multi-hop settings.
>
> |           |  n  |  TP |  TN |  FP |  FN |  Acc  | Precision | Recall |   F1  |
> |-----------|:---:|:---:|:---:|:---:|:---:|:-----:|:---------:|:------:|:-----:|
> | Zero-shot | 743 | 147 | 298 |  73 | 225 | 0.599 |   0.668   |  0.395 | 0.497 |
> |  Few-shot | 796 | 164 | 224 | 174 | 234 | 0.487 |   0.485   |  0.412 | 0.446 |
> |    CoT    | 756 | 373 | 285 |  96 |  2  | 0.870 |   0.795   |  0.995 | 0.884 |
>
> [1] Press Ofir et al. Measuring and narrowing the compositionality gap in language models. EMNLP 2023, Findings.

---

### Official Review · Reviewer_pvzs · 2023-11-01

**Soundness:** 3 good
**Presentation:** 3 good
**Contribution:** 3 good
**Rating:** 6
**Confidence:** 4

**Summary:**

This paper introduces a new evaluation framework called KNOWLEDGE CONFLICT to assess the abilities of large language models (LLMs) to handle knowledge conflicts. Knowledge conflicts arise when there is a discrepancy between the LLM's internal parametric knowledge and external non-parametric knowledge provided in the prompt context.
To evaluate this, the framework includes tasks to test LLMs on:
-Detecting contextual knowledge conflicts
-Pinpointing conflicting spans in QA settings
-Generating distinct answers drawing on conflicting knowledge

The authors find that while LLMs can identify knowledge conflicts, they struggle with localizing conflicts and producing distinct responses. New instruction-based methods are proposed that improve performance on conflict detection and distinct answer generation. The analysis also reveals factors impacting conflict handling abilities.

**Strengths:**

- The topic is an important open problem of handling knowledge conflicts in LLMs.
- Writing is clear and well-presented.
- Introduces a comprehensive evaluation framework with diverse, complex test cases

**Weaknesses:**

-Framework limited to word-level knowledge edits, more complex conflicts may be harder
The hallucination is possible in LLM's answer. It seems that this is not well addressed in the paper.

**Questions:**

- Could the assumption of a single parametric knowledge answer be relaxed? How would results change?

---

> ### Author Response · Authors · 2023-11-16
>
> We would like to thank the reviewer for their insightful comments and feedback. Their positive remarks regarding the writing, motivation, and our suggested evaluation framework are truly encouraging.
>
> > Framework limited to word-level knowledge edits, more complex conflicts may be harder The hallucination is possible in LLM's answer. It seems that this is not well addressed in the paper.
>
> We appreciate your observation regarding the framework being limited to word-level knowledge edits and the potential for hallucinations in the LLM's answer. They are good future directions and we also consider and discuss them in Appendix B.
>
> We agree that real-world knowledge conflicts might be more complex, but we found that **LLMs even fail to address these *easy* knowledge conflicts in our work generated by word-level edits**. We also admit that there is a chance of hallucination, but since our entity list is generated by LLMs (Section 2.1), it is biased towards entities already well-known to LLMs, reducing the likelihood of hallucination. Moreover, it is still debatable whether hallucinations should be considered as parametric knowledge, and our primary focus is on conflicts/knowledge differences instead of whether the parametric knowledge is factually accurate or not. The abilities of identifying the existence of knowledge conflicts, pinpointing specific conflicting segments, and generating distinct responses based on all conflicting information are the desiderata that we emphasize and evaluate in this work.
>
> > Could the assumption of a single parametric knowledge answer be relaxed? How would results change?
>
> We conduct **an additional experiment** to explore the possibility of multiple parametric knowledge answers. We sample 600 questions (with conflicts) across different knowledge domains, and employ gpt-3.5-turbo to generate 3 answers in a zero-shot manner with a temperature of 1.5 to try to induce multiple parametric knowledge answers, following self-consistency [1]. We present the results in the following table:
>
>
>
> |                       | ExactMatch | Synonyms | Invalid/Incorrect |
> |-----------------------|------------|----------|-------------------|
> | Mathematics           | 8          | 17       | 5                 |
> | Biology               | 9          | 18       | 3                 |
> | Chemistry             | 7          | 20       | 3                 |
> | Physics               | 4          | 23       | 3                 |
> | Psychology            | 6          | 20       | 4                 |
> | Computer Science      | 7          | 21       | 2                 |
> | History               | 11         | 18       | 1                 |
> | Literature            | 15         | 14       | 1                 |
> | Sociology             | 3          | 27       | 0                 |
> | Economics             | 9          | 19       | 2                 |
> | Political Science     | 10         | 16       | 4                 |
> | Philosophy            | 5          | 21       | 4                 |
> | Anthropology          | 6          | 24       | 0                 |
> | Geology               | 8          | 17       | 5                 |
> | Linguistics           | 1          | 25       | 4                 |
> | Art History           | 12         | 10       | 8                 |
> | Environmental Science | 8          | 19       | 3                 |
> | Health Sciences       | 7          | 22       | 1                 |
> | Communications        | 9          | 20       | 1                 |
> | Music                 | 3          | 20       | 7                 |
> | **Total**             | **148**    | **391**  | **61**            |
>
> The results show that 25% of the additional parametric answers matched the top answer exactly with 65% of synonyms, while others tend to be invalid or incorrect answers. Therefore, we conclude that the vast majority of questions have only one parametric answer, thus the results and findings won’t change significantly if we relax this assumption.
>
> [1] Wang, Xuezhi, et al. Self-Consistency Improves Chain of Thought Reasoning in Language Models. ICLR 2022.

---

### Official Review · Reviewer_o2Z4 · 2023-11-08

**Soundness:** 2 fair
**Presentation:** 3 good
**Contribution:** 2 fair
**Rating:** 5
**Confidence:** 4

**Summary:**

The authors propose a framework to evaluate LLMs’ ability to handle knowledge conflicts, which includes: 1) identifying contextual knowledge conflicts, 2) pinpointing conflicting knowledge segments, and 3) providing distinct answers or viewpoints amidst conflicts. Under the setting the authors proposed above, the instruction-based approach is introduced to alleviate these problems.

**Strengths:**

1.	This article breaks down the evaluation aspects of knowledge conflict issues in a fine-grained manner and proposes a reasonable idea that LLMs should not rely solely on either parametric or non-parametric information, but grant LLM users the agency to make informed decisions based on distinct answers.
2.	For the three proposed tasks, this paper designed plenty of experiments for verification. The motivation is clear and the prompt templates are straightforward.

**Weaknesses:**

1.	The experimental settings are not rigorous. The data sets corresponding to the three knowledge conflict tasks are generated according to several rules (entity substitution and shuffling), and then the proposed approaches (prompt templates) are strongly related to these artificial rules. That is my main concern: with those settings, the experiments in the paper might have limited value and provide limited insights. Besides, this paper seems to lack a connection to previous works in the field of knowledge conflict. The organization of the entire article is isolated and does not introduce other benchmarks/analysis[1] or other method comparisons that specifically address knowledge conflict issues.
2.	The limited size of the knowledge conflict dataset proposed in this paper makes the analysis unconvincing. Take Figure 4 as an example, the authors argue that the ability to tackle knowledge conflict varies across knowledge domains. However, according to Table 7 in the appendix, on average there are only about 100 test cases per domain, which I think is far from enough to claim that knowledge conflict varies across knowledge domains.

[1] DisentQA: Disentangling Parametric and Contextual Knowledge with Counterfactual Question Answering. ACL 2023

**Questions:**

No more questions.

---

> ### Author Response · Authors · 2023-11-16
>
> We would like to thank the reviewer for their thoughtful comments and feedback. We are encouraged by their positive feedback on the motivation of our work, as well as our experiments for verification.
>
> > The experimental settings are not rigorous. The data sets corresponding to the three knowledge conflict tasks are generated according to several rules (entity substitution and shuffling), and then the proposed approaches (prompt templates) are strongly related to these artificial rules. That is my main concern: with those settings, the experiments in the paper might have limited value and provide limited insights.
>
> In response to the question of why we don't perceive our work as limited, it's essential to highlight that our contributions extend beyond the synthetic datasets and proposed approaches.
> The abilities of 1) *identifying the existence of knowledge conflicts*, 2) *pinpointing specific conflicting segments*, and 3) *generating distinct responses based on all conflicting information* are the desiderata that we emphasize and evaluate in this work.
>
> To this end, the proposed instruction-based methods for prompting and fine-tuning are indeed related to the three objectives we champion, but none of them depend on specific conflict generation methods (entity substitution and shuffling). What’s important is that: **even with these specifically tailored approaches, LLM performance is far from perfect across all three tasks**, which indicates that resolving knowledge conflicts is a hard problem and sheds light on the need for future work.
>
> In terms of the datasets, we propose an evaluation framework to generate synthetic datasets instead of adopting existing datasets since the issue of knowledge conflicts heavily depends on the unique parametric knowledge of each LLM, so we can’t safely assume any existing datasets as the parametric knowledge. Moreover, existing open-domain QA datasets may not be well-suited for examining contextual knowledge conflicts, and other works [1-2] on knowledge conflicts that use existing datasets employ synthetically generated knowledge conflicts as well.
>
> [1] Shayne Longpre, Kartik Perisetla, Anthony Chen, Nikhil Ramesh, Chris DuBois, and Sameer Singh. Entity-based knowledge conflicts in question answering. EMNLP 2021.
>
> [2] Jian Xie, Kai Zhang, Jiangjie Chen, Renze Lou, and Yu Su. Adaptive chameleon or stubborn sloth: Unraveling the behavior of large language models in knowledge clashes. ArXiv, abs/2305.13300, 2023.

---

> ### Author Response · Authors · 2023-11-16
>
> > Besides, this paper seems to lack a connection to previous works in the field of knowledge conflict. The organization of the entire article is isolated and does not introduce other benchmarks/analysis[1] or other method comparisons that specifically address knowledge conflict issues.
> >
> > [1] DisentQA: Disentangling Parametric and Contextual Knowledge with Counterfactual Question Answering. ACL 2023
>
> Thank you for the pointer to DisentQA, which explores and analyzes the ability to disentangle parametric and contextual knowledge, and it is relevant to Task 3 in our study. DisentQA focuses on training data augmentation methods and relies on Natural Questions, while we focus on instruction-based methods and employ synthetic datasets generated by our proposed framework, as NQ has been reasonably addressed in existing works [1-2] and we want to refrain from assuming existing datasets as the parametric knowledge. We further introduce the entity shuffling approach which addresses the limitation of their counterfactual data augmentation process which can only be employed for questions whose answers are named entities, and break down experimental results by different knowledge domains and conflict generation methods. While there are differences between our work and DisentQA, it is a valuable reference and we will add it to the related work in the final version.
>
> While there are several works on simulating knowledge conflicts and analyzing the behavior of LLMs in the face of knowledge conflicts (as we cited in the paper), including DisentQA, we are the first to define the desiderata and propose concrete objectives towards resolving knowledge conflicts.
>
> > The limited size of the knowledge conflict dataset proposed in this paper makes the analysis unconvincing. Take Figure 4 as an example, the authors argue that the ability to tackle knowledge conflict varies across knowledge domains. However, according to Table 7 in the appendix, on average there are only about 100 test cases per domain, which I think is far from enough to claim that knowledge conflict varies across knowledge domains.
>
> As shown in Table 7, we employ approximately 450 test cases per knowledge domain: the "Total" column is the amount of test cases in each knowledge domain. With 20 distinct knowledge domains, there are a total of 9,000 test cases in the experiments. This quantity should be sufficient for drawing meaningful conclusions and is higher than in similar works that study knowledge conflicts [3-4].
>
> [1] Zhang, Hang et al. Poolingformer: Long Document Modeling with Pooling Attention. ICML, 2021.
>
> [2] Xuguang Wang et al. No Answer is Better Than Wrong Answer: A Reflection Model for Document Level Machine Reading Comprehension. EMNLP 2020, Findings.
>
> [3] Jian Xie, Kai Zhang, Jiangjie Chen, Renze Lou, and Yu Su. Adaptive chameleon or stubborn sloth: Unraveling the behavior of large language models in knowledge clashes. ArXiv, abs/2305.13300, 2023.
>
> [4] Neeman Ella et al. DisentQA: Disentangling Parametric and Contextual Knowledge with Counterfactual Question Answering. ACL 2023.

---

### Author Response · Authors · 2023-11-20
**Revised Paper Posted**

Dear reviewers,

We appreciate your valuable comments and feedback. We have integrated all the edits you suggested and released an updated version, with updates including citing and discussing suggested references, presenting newly added experiments and results, and offering a more in-depth analysis and discussion of the results and future prospects. Any further feedback you may have would be greatly appreciated.

Thank you,
authors

---

### Author Response · Authors · 2023-11-22
**Looking for Final Review and Feedback**

Dear reviewers,

We would like to once again convey our sincere gratitude for the valuable feedback you provided. Your insights have been incredibly helpful and meaningful. As the discussion period is concluding, we kindly wonder if you can take a moment to review our responses and updates, which we believe have been significantly strengthened with your valuable suggestions. Also feel free to let us know if you have any further questions.

Thank you,
authors

---

### Meta-Review · Area_Chair_ixvM · 2023-12-10

**Metareview:**

The authors present a comprehensive framework for assessing the capability of large language models (LLMs) to manage knowledge conflicts. This framework consists of three key components: identifying contextual knowledge conflicts, pinpointing specific segments of conflicting information, and providing distinct responses or perspectives in the presence of disagreements. To address these challenges, the authors propose an instruction-based approach within the context of their established framework. The topic is of interest to the community.

A major concern was raised regarding the relationship between this paper and the related work DisentQA. The question is whether this paper offers any new significant insights beyond those already discussed in DisentQA, which has explored how to probe knowledge conflict and generate distinct answers by disentangling knowledge. The area chair concurs with the reviewer's concerns, and it appears that the specific method proposed to address the issue of knowledge conflict heavily depends on the paper's objectives, which lack sufficient generalizability.

**Justification For Why Not Higher Score:**

A major concern was raised regarding the relationship between this paper and the related work DisentQA. The question is whether this paper offers any new significant insights beyond those already discussed in DisentQA, which has explored how to probe knowledge conflict and generate distinct answers by disentangling knowledge. The area chair concurs with the reviewer's concerns, and it appears that the specific method proposed to address the issue of knowledge conflict heavily depends on the paper's objectives, which lack sufficient generalizability.

**Justification For Why Not Lower Score:**

N/A

---

### Decision · Program_Chairs · 2024-01-16

Reject